# World-Model Inspired Emotion-aware Token Refinement for Training-Free Multimodal Emotion Recognition

Kejun Liu [1]   Zhe Chen [2]   Yuanyuan Liu [* 1]   ke Wang [1]   Yibing Zhan [3]   Wei Xiang [2]   Hongyan Zhang [1]

## Abstract

Multimodal Large Language Models (MLLMs) show promise for Multimodal Emotion Recognition (MER) but often remain unreliable because sparse emotional cues could be easily overwhelmed and affected by redundant context. While fine-tuning is effective, it is usually costly when using large models. Training-free methods like chain-of-thought reasoning provide a practical alternative, but they mostly rely on heuristic prompting to influence the model behaviors and do not explicitly focus on emotion relevant tokens internally, which would allow decision-relevant emotional tokens to be diluted by environmental noise, resulting in unstable predictions. To address this limitation without training, we rethink MER from a world-model perspective that treats emotion as a latent state inferred from noisy and redundant multimodal observations. Under frozen parameters, this view suggests that robustness depends on constraining why and how tokens contribute to inference. Based on this insight, we propose **WETR** (**W**orld-Model inspired **E**motion-aware **T**oken **R**efinement), a training-free, plug-and-play regulator that reshapes token usage through two mechanisms: Noise-suppressed Token Selection (NTS), which suppresses redundant intra-modal noise, and State-strengthened Token Reweighting (STR), which amplifies decision-relevant emotional tokens. Experiments on multiple MER benchmarks demonstrate that WETR consistently improves accuracy and stability under frozen parameters, which also improves token-level interpretability.

[1]School of Computer Science, China University of Geosciences (Wuhan), Wuhan, China [2]School of Computing, Engineering and Mathematical Sciences, La Trobe University, Melbourne, Australia [3]School of Computer Science, Wuhan University, Wuhan, China. Correspondence to: **\*Yuanyuan Liu (Corresponding author)** <liuyy@cug.edu.cn>.

*Proceedings of the $43^{rd}$ International Conference on Machine Learning*, Seoul, South Korea. PMLR 306, 2026. Copyright 2026 by the author(s).

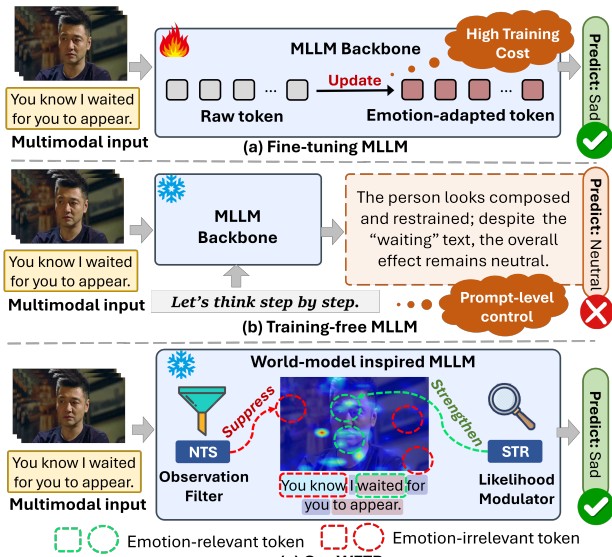

*Figure 1.* **Different methods to improve MER with MLLMs.** (a) Fine-tuning updates parameters to obtain emotion-adapted tokens. (b) Training-free prompting may be distracted by emotion-irrelevant context (predict neutral). (c) WETR refines tokens at inference by suppressing irrelevant tokens (NTS) and strengthening emotion cues (STR), correcting to sad.

## 1. Introduction

Multimodal large language models (MLLMs) have attracted increasing attention for their strong cross-modal understanding and reasoning abilities (Li et al., 2024; Chen et al., 2024b; Liu et al., 2024; Huang et al., 2024), achieving impressive performance on tasks such as visual question answering and visual reasoning (Zhang et al., 2025). Despite this progress, applying MLLMs to multimodal emotion recognition (MER) remains challenging: emotional expressions are often subtle and sparsely distributed (*e.g.*, small facial muscle changes in videos, contrastive turns in text, etc..) (Cheng et al., 2024; Yang et al., 2024), while real-world data typically contains substantial redundant and irrelevant context or background noise. As a result, MLLMs may struggle to attend to decision-relevant emotional cues. Consequently, even if MLLMs could generate a seemingly reasonable emotional response, its decision may not be grounded in the truly informative emotional evidence, leading to unstable and unreliable emotion reasoning.

Recent studies have begun to address the difficulty MLLMs face in leveraging decision-relevant emotional cues for MER (Zhang et al., 2024a; Zhao et al., 2024; Zhang et al., 2024b). Existing approaches largely fall into two categories: fine-tuning-based methods and training-free inference enhancements. Fine-tuning approaches typically train specialized encoders to better capture emotional signals across modalities (Cheng et al., 2024), or employ auxiliary MLLMs to generate emotion-aware textual descriptions (Xie et al., 2024), followed by supervised training on emotion-related datasets. While effective, such methods incur substantial training costs, limit scalability, and are often impractical in realistic deployments that depends on frozen large-scale MLLMs. Motivated by these constraints, an increasing number of studies explore training-free inference-time strategies, including prompt engineering, chain-of-thought reasoning, and self-reflection (Zhang et al., 2024a; Fang et al., 2025). Although these methods can guide MLLMs toward emotion-related information, they primarily operate at the input or output level and continue to rely on the pretrained token-level attention and aggregation mechanisms. Without explicitly regulating how decision-relevant emotional tokens are used during inference, predictions could remain unstable when irrelevant context or background redundancy dominates. Fig. 1 provides a conceptual comparison between fine-tuning, training-free prompting, and our inference-time token refinement for MER. This leads to a fundamental question: *Is it possible to stabilize MER without retraining by explicitly steering inference toward decision-relevant emotional tokens?*

In this work, we rethink the role of decision-relevant emotional tokens in MER from a world-model perspective, motivated by the observation that emotion recognition can be inherently a problem of latent state inference rather than direct pattern matching. Rather than direct observations from multimoadl inputs, we hypothesize that emotional states manifest through sparse, fine-grained cues, such as subtle facial muscle movements or contrastive turns in text, embedded within substantial irrelevant or redundant context. Accordingly, we formulate MER as inferring a latent emotional world state from multimodal observations that mix informative emotional evidence with modality-specific noise. Under this formulation, when MLLMs are used with frozen parameters, directly disentangling emotional signals from noise by modifying model representations could be infeasible. Robust inference thus depends on structuring how observed evidence is used, rather than learning new representations. In MLLMs, multimodal inputs are processed only through tokens, which serve as the interface for internal reasoning. From a world-model perspective, only tokens that meaningfully constrain the latent emotional state should dominate inference, while noise-dominated tokens should be suppressed. This task-induced representation makes explicit what it means for evidence to be relevant, and motivates controlling token participation and influence during inference as a principled mechanism for robust MER, forming the basis of our method.

Driven by the world-model assumption that robust emotion recognition requires prioritizing evidence that constrains the latent state, we propose **WETR** (World-Model Inspired Emotion-aware Token Refinement), a training-free, plug-and-play inference regulator for frozen MLLMs. WETR consists of two complementary components that bridge raw observations to the latent emotional state: (1) **Noise-suppressed Token Selection (NTS)** acts as an **observation filter**, isolating a compact set of emotion-relevant tokens from high-dimensional input; this reflects the world-model premise that only sparse, task-relevant observations meaningfully reduce uncertainty about the latent emotional state. (2) **State-strengthened Token Reweighting (STR)** serves as a **likelihood modulator**, adjusting token influence during multimodal aggregation to emphasize decision-relevant emotional tokens that provide the strongest constraints on the latent state while down-weighting uninformative tokens. By iteratively refining token these components at sparse trigger layers, WETR encourages inference to focus on state-constraining emotional evidence while suppressing noise, yielding more stable predictions and better token-level interpretability without parameter updates. **Our main contributions are summarized as follows:**

- We propose WETR, a training-free, plug-and-play inference-time regulator for MLLMs, inspired by a world-model perspective to explicitly refine decision-relevant emotional token usage for robust MER under noisy context.

- We introduce Noise-suppressed Token Selection (NTS), which selects a compact emotion-relevant set to suppress intra-modal redundancy while preserving decision-relevant emotional cues.

- We design State-strengthened Token Reweighting (STR), which reweights retained tokens to strengthen complementary cues and reduce the influence of weakly informative tokens.

- We validate WETR on MAFW, CH-SIMS, and MER23, demonstrating consistent improvements in accuracy and stability.

## 2. Related Work

### 2.1. MLLMs-based MER

Large Multimodal Language Models (MLLMs) have advanced MER by enabling unified, instruction-following reasoning across modalities via shared language representa-

tions (Yang et al., 2025; Liang et al., 2025). Emotion-LLaMA (Cheng et al., 2024) designed some emotion-specific encoders and employed instruction tuning to align and fuse multimodal features, significantly enhancing the ability of emotion recognition and reasoning. Affect-GPT (Lian et al., 2025) constructed the largest fine-grained descriptive emotion dataset MER-Caption and a unified benchmark MER-UniBench specifically designed for evaluating the MER tasks of MLLMs. R1-Omni (Zhao et al., 2025) applied Reinforcement Learning with Verifiable Reward (RLVR) to multimodal emotion understanding tasks for the first time, significantly enhancing the generalization ability, reasoning ability and interpretability of MLLMs in out-of-distribution scenarios. Despite strong understanding, these methods require fine-tuning, leading to high training costs. We aim to enhance the emotional recognition capabilities of MLLMs during inference, without relying on extra training, making emotional understanding more scalable.

## 2.2. Training-Free MLLMs

Due to the high training costs associated with MLLMs, training-free methods have gained considerable attention as a fresh perspective (Bi et al., 2025; Kim et al., 2024a; Chen et al., 2024a; Kim et al., 2024b), particularly in tasks such as inference acceleration and mitigating MLLM hallucinations. Typical training-free methods such as text prompt engineering to guide model behavior and improve outcomes without additional training, token merging to enhance processing efficiency by simplifying input data, and visual augmentation alongside multi-step inference to boost performance through visual enhancements and iterative processing. In emotion-related tasks, SEPM (Fang et al., 2025) uses confidence-guided coarse-to-fine prompting with a Focus-on-Emotion query to drop emotion-irrelevant visual tokens, improving recognition. However, most training-free strategies act on the input prompting or compress tokens generically, offering limited control over token usage. In contrast, we target inference-time token usage directly by refining both token admission and reweighting to stabilize MER under noisy context.

## 3. The Proposed Method

### 3.1. Problem Definition

Given a video-text pair $(V, T)$ where $V$ is a sequence of frames and $T$ is the associated text, our goal is to predict an emotion state $\hat{y}$ via frozen MLLMs. A frozen MLLM serializes $(V, T)$ into a unified token sequence $\mathbf{x} = [x_1, \ldots, x_L]$, where $x_i$ is the $i$-th token and $L$ is the sequence length. We use index sets to distinguish token types: $\mathcal{I}_v \subset \{1, \ldots, L\}$ for visual tokens and $\mathcal{I}_t \subset \{1, \ldots, L\}$ for text tokens. To avoid template tokens contaminating statistics, we further split text tokens into content tokens $\mathcal{I}_{con} \subseteq \mathcal{I}_t$ and instruc-

tion tokens $\mathcal{I}_{inst} \subseteq \mathcal{I}_t$, where $\mathcal{I}_{inst}$ is always preserved to keep prompting stable and outputs parsable.

### 3.2. Token Usage under World-Model View

**World-Model Formulation for MER**   To investigate how decision-relevant emotional tokens are used during inference, we adopt a world-model view of MER. We treat emotion as a latent emotional world state $z_e$ that gives rise to multimodal observations, while each modality also contains noise factors that are irrelevant to emotion. Concretely, we abstract a video-text pair $(V, T)$ as

$$V = g_v(z_e, \varepsilon_v), \qquad T = g_t(z_e, \varepsilon_t), \tag{1}$$

where $\varepsilon_v$ and $\varepsilon_t$ capture modality-specific noise (e.g., background redundant, irrelevant context), and $g_v(\cdot), g_t(\cdot)$ are generative mappings. This formulation does not assume access to $g_v, g_t$; it is used to clarify the inference goal: robust MER should reduce uncertainty about $z_e$ using sparse emotion-relevant cues while resisting $\varepsilon$-dominated redundancy. When MLLMs are used with frozen parameters, directly learning new representations to disentangle $z_e$ from $\varepsilon$ is infeasible. Therefore, robustness must come from *structuring how observed evidence is used* during inference, rather than modifying representations via training. Since MLLMs process multimodal inputs purely through tokens, this world-model view points to a principled and controllable lever: *token usage*.

**Token Usage as the Controllable Inference Lever**   Let $\mathbf{x} = [x_1, \ldots, x_L]$ be the unified token sequence serialized from $(V, T)$. Under frozen parameters, our intervention is restricted to inference-time token participation and influence. We therefore introduce a token-usage weight vector $w = \{w_i\}_{i=1}^L$, where $w_i$ controls how strongly token $x_i$ contributes to the model's internal aggregation for inferring $z_e$. We factorize token usage as

$$w_i = m_i \gamma_i, \tag{2}$$

where $m_i \in \{0, 1\}$ is a **selection factor** that decides whether token $i$ is admitted as evidence, and $\gamma_i > 0$ is a **reweighting factor** that modulates the influence of admitted tokens. We denote the admitted set as $\mathcal{K} = \{i \mid m_i = 1\}$.

**Our WETR.**   We therefore propose **WETR** (**W**orld-Model inspired **E**motion-aware **T**oken **R**efinement), a plug-and-play inference-time regulator that sparsely updates the two factors of $w_i$ at a set of trigger layers. Specifically, at each trigger layer (as shown in Fig. 2), **NTS** updates $m_i$ and acts as an **observation filter**, selecting a compact set of emotion-relevant tokens to suppress intra-modality redundancy; **STR** updates $\gamma_i$ and acts as a **likelihood modulator**, re-scaling the effective likelihood contribution of retained tokens to down-weight predictable, weakly emotion-relevant

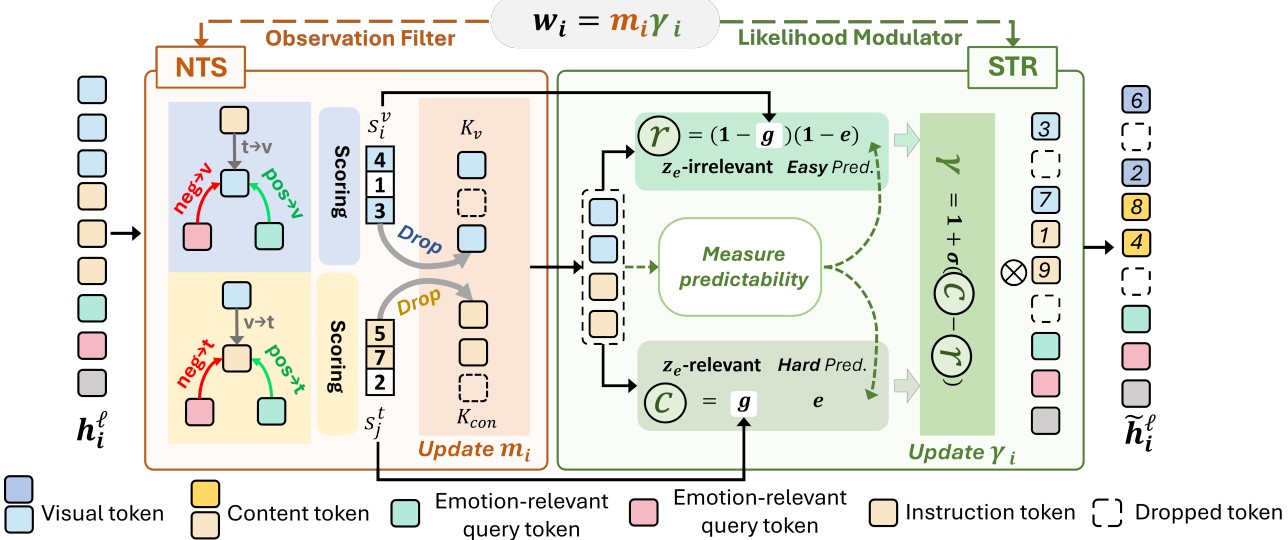

*Figure 2.* **Our WETR at trigger layer** $\ell$. Given the token hidden states $\{\mathbf{h}_i^\ell\}$, WETR refines token usage $w_i = m_i\gamma_i$ via two modules. **NTS** (observation filter) scores visual/content tokens using contrastive emotion queries and updates the selection factor $m_i$ to keep a compact emotion-relevant set $\mathcal{K}$. **STR** (likelihood modulator) measures cross-modal predictability within $\mathcal{K}$, then updates the reweighting factor $\gamma_i$ to down-weight predictable, weakly emotion-relevant overlap and up-weight hard-to-predict emotion cues. The refined hidden states $\tilde{\mathbf{h}}_i^\ell = \gamma_i\mathbf{h}_i^\ell$ are passed to subsequent layers, while non-trigger layers remain unchanged.

overlap and up-weight emotion-relevant residual cues that provide larger incremental constraints on $z_e$. We define the trigger set as $\mathcal{L}_{reg} = \{\ell \mid \ell \bmod n_r = 0\}$ and leave all non-trigger layers unchanged. The final prediction $\hat{y}$ is produced by standard decoding on the refined hidden states. Overall, this sparse refinement progressively shifts inference away from $\varepsilon$-dominated noisy tokens toward tokens that better constrain $z_e$, improving stability without any parameter updates.

### 3.3. NTS: Noise-suppressed Token Selection

**Goal.** As the **observation filter** in WETR, NTS updates the **selection factor** $m_i$ in $w_i = m_i\gamma_i$ to control token admission under frozen parameters. It constructs a compact emotion-relevant evidence set $\mathcal{K}$ that suppresses intra-modality redundancy (tokens largely explained by $\varepsilon$) while retaining sparse cues that are most informative for inferring the latent emotion state $z_e$. Formally, the desired utility of admitting token $x_i$ is its conditional information gain,

$$\Delta_i^\star = I(z_e; x_i \mid \mathbf{x}_{-i}), \qquad (3)$$

which measures how much $x_i$ reduces uncertainty about $z_e$ beyond what is already explained by the remaining tokens. Since $\Delta_i^\star$ is not directly accessible in frozen MLLMs, NTS estimates a computable proxy $\widehat{\Delta}_i$ during inference and admits the top-scoring tokens into $\mathcal{K}$.

**Contrastive-query utility proxy.** To obtain a computable proxy for the conditional gain in Eq. (3), we score each

token by *contrastive querying*. We insert two short instruction spans: an **emotion-relevant** query $Q_{\text{pos}}$ that explicitly seeks decision-relevant emotional cues, and an **emotion-irrelevant** query $Q_{\text{neg}}$ that can be answered without emotion, thus reflecting generic context usage. Let their spans be $\mathcal{I}_{\text{pos}}, \mathcal{I}_{\text{neg}} \subseteq \mathcal{I}_{inst}$. At trigger layer $\ell$, with head-aggregated attention $\mathbf{A}^{(\ell)}$, denote by $a_{\mathcal{I} \rightarrow i}^{(\ell)}$ the average attention mass from span $\mathcal{I}$ to token $i$. We first compute a contrastive utility

$$d_i = \text{Norm}\left(a_{\mathcal{I}_{\text{pos}} \rightarrow i}^{(\ell)} - a_{\mathcal{I}_{\text{neg}} \rightarrow i}^{(\ell)}\right). \qquad (4)$$

We use the difference rather than $a_{\mathcal{I}_{\text{pos}} \rightarrow i}^{(\ell)}$ alone to cancel query-shared, $\varepsilon$-driven attention and prompt-induced global bias, so $d_i$ better reflects the incremental evidence of $x_i$ for constraining $z_e$ in Eq. (3). As a result, tokens mainly explained by irrelevant context tend to receive similar attention from both queries and are suppressed after subtraction, while emotion-informative tokens remain salient. To reduce modality bias, we add a weak cross-modal support term and define the final selection score:

$$\widehat{\Delta}_i = s_i = d_i + \begin{cases} \text{Norm}\left(\frac{1}{|\mathcal{I}_{\text{con}}|} \sum_{q \in \mathcal{I}_{\text{con}}} \mathbf{A}_{q,i}^{(\ell)}\right), & i \in \mathcal{I}_v, \\ \text{Norm}\left(\frac{1}{|\mathcal{I}_v|} \sum_{q \in \mathcal{I}_v} \mathbf{A}_{q,i}^{(\ell)}\right), & i \in \mathcal{I}_{\text{con}}. \end{cases}$$
$$(5)$$

We use $s_i$ as the proxy utility $\widehat{\Delta}_i$ for ranking tokens toward the information-gain objective $\Delta_i^\star$.

**Top-$p$ emotion-relevant set construction.** Using the final scores $s_i$, NTS admits only the highest-utility tokens to form a compact emotion-relevant set. To prevent a single

modality from dominating, we apply top-$p$ filtering within each modality: we keep a ratio $p_v$ of visual tokens and a ratio $p_t$ of text-content tokens. Let $\text{TopP}(\cdot, p)$ return the indices of the top-$p$ fraction ranked by scores; we obtain

$$\mathcal{K}_v = \text{TopP}(\{s_i\}_{i \in \mathcal{I}_v}, p_v), \quad \mathcal{K}_{\text{con}} = \text{TopP}(\{s_j\}_{j \in \mathcal{I}_{\text{con}}}, p_t). \tag{6}$$

The final emotion-relevant set is $\mathcal{K} = \mathcal{K}_v \cup \mathcal{K}_{\text{con}} \cup \mathcal{I}_{\text{inst}}$, where $\mathcal{I}_{\text{inst}}$ is always preserved for prompt stability and parsable outputs. Accordingly, the selection factor is

$$m_i = \mathbb{1}[i \in \mathcal{K}], \tag{7}$$

where $\mathbb{1}[\cdot]$ is the indicator function.

### 3.4. STR: State-strengthened Token Reweighting

**Goal.** As the **likelihood modulator** in WETR, STR updates the **reweighting factor** $\gamma_i$ in $w_i = m_i \gamma_i$ to control how strongly retained tokens contribute to aggregating evidence for the latent emotion state $z_e$. Under the world-model view, each retained token can be regarded as providing a likelihood term $\log p(x_i \mid z_e)$; reweighting $\gamma_i$ corresponds to modulating its effective likelihood contribution, i.e.,

$$\log \tilde{p}(z_e \mid \mathbf{x}; w) \propto \log p(z_e) + \sum_{i \in \mathcal{K}} (m_i \gamma_i) \log p(x_i \mid z_e). \tag{8}$$

Since $\log p(x_i \mid z_e)$ is not directly accessible in frozen MLLMs, STR aims to redistribute $\gamma_i$ within $\mathcal{K}$ so that tokens with higher incremental utility for constraining $z_e$ receive larger weights, while easy-to-match, $\varepsilon$-driven overlap with low marginal gain is down-weighted. For stability, STR is gated by the NTS score and preserves a small set of high-confidence anchors.

Let $\mathbf{H} \in \mathbb{R}^{L \times d}$ be the hidden states at trigger layer $\ell$ (token $i$: $\mathbf{h}_i$), and let $\mathbf{q}_i, \mathbf{k}_i$ denote its query/key vectors from the frozen attention projections.

**Identifying replaceable overlap vs. complementary residual evidence.** To realize the likelihood modulation in Eq. (8) without training, STR needs a computable cue to redistribute $\gamma_i$ within $\mathcal{K}$. We distinguish (i) *replaceable overlap* that is largely predictable from the other modality under the current alignment, from (ii) *complementary residual evidence* that is hard to predict and thus can provide larger incremental constraints on $z_e$. This does not dismiss agreement: if $x_i$ is already well explained by $\mathbf{x}_{-i}$, its conditional gain $I(z_e; x_i \mid \mathbf{x}_{-i})$ is typically small, so increasing its likelihood weight yields limited posterior tightening. Accordingly, STR mainly down-weights predictable but weakly emotion-relevant tokens, keeps emotion-relevant agreement largely unchanged, and up-weights emotion-relevant residual cues.

We use cross-modal predictability as a lightweight indicator of replaceability. Under the query-induced alignment at

layer $\ell$, tokens whose keys are well reconstructable from the other modality are treated as overlap, while poorly reconstructable ones are treated as residual evidence. For $i \in \mathcal{K}_v$, we compute

$$\alpha_{ij} = \text{Softmax}_{j \in \mathcal{K}_{con}} \left( \frac{\mathbf{q}_i^\top \mathbf{k}_j}{\sqrt{d}} \right), \tag{9}$$

and measure predictability by the reconstruction error $e_i^v = \left\| \mathbf{k}_i - \sum_{j \in \mathcal{K}_{con}} \alpha_{ij} \mathbf{k}_j \right\|_2$. We define $e_j^t$ symmetrically (reconstructing from $\mathcal{K}_v$) and normalize all errors to $\bar{e}_i \in [0, 1]$. Predictability alone does not indicate emotion relevance, so we gate it by the NTS score $s_i$: $g_i = \varphi(\text{Norm}(s_i)) \in [0, 1]$, where $\varphi(\cdot)$ is a sigmoid gate. We then define

$$r_i = (1 - g_i)(1 - \bar{e}_i), \qquad c_i = g_i \bar{e}_i, \tag{10}$$

where $r_i$ highlights predictable but weakly emotion-relevant overlap, while $c_i$ highlights emotion-relevant yet hard-to-predict residual evidence. Notably, emotion-relevant agreement corresponds to high $g_i$ and low $\bar{e}_i$, yielding small $r_i$ and $c_i$, so it is not down-weighted ($\gamma_i \approx 1$).

**Reweighting update with anchor protection.** Based on the overlap/residual scores $(r_i, c_i)$, STR updates $\gamma_i$ to implement the likelihood modulation in Eq. (8): it down-weights predictable but weakly emotion-relevant overlap (large $r_i$) and up-weights emotion-relevant residual evidence (large $c_i$). We use a signed migration bias

$$b_i = c_i - r_i, \qquad \gamma_i = 1 + \delta b_i, \tag{11}$$

where $\delta > 0$ controls the intervention strength. To avoid over-suppressing reliable evidence, we protect a small anchor set $\mathcal{A}$ selected by NTS scores (top-$p_v^{anc}$ within $\mathcal{K}_v$ and top-$p_t^{anc}$ within $\mathcal{K}_{con}$). For anchors, we enforce a lower bound by clamping $\gamma_i \leftarrow \max(\gamma_i, \gamma_{anc})$, where $\gamma_{anc} \geq 1$. Finally, we apply token-wise scaling $\widetilde{\mathbf{h}}_i = \gamma_i \mathbf{h}_i$ so subsequent layers attend on $\widetilde{\mathbf{h}}_i$, realizing the refined usage $w_i = m_i \gamma_i$.

## 4. Experiment

We design experiments to answer the following questions:

**RQ1:** Does WETR consistently improve MER across frozen MLLMs under various datasets? (Sec. 4.3)

**RQ2:** Can NTS select compact emotion-relevant evidence by suppressing $\varepsilon$-dominated tokens? (Sec. 4.4)

**RQ3:** Can STR reweight retained tokens to strengthen state-informative emotional cues? (Sec. 4.4)

**RQ4:** Under noise amplification ($\varepsilon \uparrow$), can WETR maintain $z_e$-constrained token usage? (Sec. 4.4)

*Table 1.* Comparison with state-of-the-art on the MAFW, CH-SIMS and MER23 database. The best results are in **bold**.

| Methods | Speed | MAFW dataset | | | SIMS dataset | | | MER23 dataset | | |
|---|---|---|---|---|---|---|---|---|---|---|
| | | WAR | UAR | F1 | ACC-2 | ACC-3 | F1 | WAR | UAR | F1 |
| Qwen2-VL-7B (Wang et al., 2024) | 5.4010s | 43.58 | 36.82 | 47.68 | 76.55 | 48.49 | 75.68 | 57.66 | 54.05 | 60.78 |
| Janus-Pro-1B (Chen et al., 2025a) | 0.5730s | 39.36 | 24.28 | 32.30 | 61.08 | 60.83 | 61.83 | 44.04 | 43.87 | 44.34 |
| InterVL2.5-2B (Chen et al., 2025b) | 0.8791s | 52.76 | 40.66 | 53.75 | 65.98 | 54.05 | 74.44 | 44.28 | 41.95 | 46.02 |
| InterVL2.5-4B (Chen et al., 2025b) | 1.1857s | 49.55 | 39.67 | 50.51 | 86.08 | 66.30 | 86.20 | 63.26 | 55.32 | 62.46 |
| InterVL3-8B (Zhu et al., 2025) | 1.0287s | 53.54 | 42.28 | 52.47 | 86.86 | 62.14 | 87.00 | 56.69 | 52.80 | 56.18 |
| HumanOmni-0.5B (Zhao et al., 2025) | – | 20.18 | 13.52 | – | – | – | – | – | – | – |
| EMER-SFT (Zhao et al., 2025) | – | 38.39 | 28.02 | – | – | – | – | – | – | – |
| PandaGPT (Su et al., 2023) | – | – | – | – | 62.37 | – | 62.93 | 39.13 | – | – |
| Emotion-LLaMA (Cheng et al., 2024) | – | – | – | – | 78.32 | – | 78.61 | 59.38 | – | – |
| MoSE (Han et al., 2025) | 2.1169s | 33.66 | 26.85 | 33.04 | 69.07 | 52.74 | 62.13 | 51.84 | 49.85 | 57.24 |
| *InterVL3.5-4B (Wang et al., 2025)* | | | | | | | | | | |
|   Zero-shot | 1.4583s | 57.76 | 45.46 | 54.73 | 86.60 | 65.21 | 86.60 | 60.83 | 59.84 | 62.03 |
|   Zero-shot-CoT (Kojima et al., 2022) | 10.2317s | 57.62 | 43.19 | 55.40 | 86.51 | 64.30 | 86.74 | 57.42 | 56.54 | 58.33 |
|   SparseVLM (Zhang et al., 2024c) | 1.9884s | 55.36 | 42.67 | 53.27 | 85.15 | 62.36 | 85.50 | 61.80 | 58.79 | 62.83 |
|   SEPM (Fang et al., 2025) | 6.0915s | 56.45 | 44.12 | 54.11 | 84.46 | 57.91 | 84.72 | 59.12 | 58.19 | 60.73 |
|   **Ours** | 1.3487s | **60.96** | **46.95** | **57.08** | **87.63** | **69.15** | **87.68** | **64.72** | **59.86** | **65.17** |
| *InterVL3.5-8B (Wang et al., 2025)* | | | | | | | | | | |
|   Zero-shot | 1.7089s | 55.08 | 44.21 | 54.19 | 86.34 | 62.36 | 86.49 | 60.34 | 57.36 | 60.57 |
|   Zero-shot-CoT (Kojima et al., 2022) | 11.0172s | 55.20 | 45.20 | 55.19 | 86.86 | 67.18 | 87.00 | 57.66 | 54.01 | 58.19 |
|   SparseVLM (Zhang et al., 2024c) | 1.7310s | 55.32 | 44.39 | 55.01 | 86.18 | 65.21 | 86.80 | 62.53 | 58.14 | 63.29 |
|   SEPM (Fang et al., 2025) | 4.8369s | 56.15 | 45.37 | 56.83 | 87.25 | 68.88 | 87.18 | 63.26 | 57.48 | 62.73 |
|   **Ours** | 1.6573s | **59.64** | **46.30** | **57.57** | **89.18** | **70.90** | **89.16** | **68.13** | **63.24** | **68.48** |
| *Qwen3-VL-4B (Bai et al., 2025)* | | | | | | | | | | |
|   Zero-shot | 2.2669s | 50.08 | 42.03 | 50.21 | 85.31 | 68.71 | 85.22 | 56.93 | 56.19 | 58.22 |
|   Zero-shot-CoT (Kojima et al., 2022) | 8.1134s | 49.66 | 40.01 | 52.38 | 86.02 | 70.51 | 85.98 | 57.66 | 56.66 | 59.05 |
|   SparseVLM (Zhang et al., 2024c) | 3.211s | 50.39 | 41.15 | 50.30 | 86.68 | 74.84 | 86.71 | 57.91 | 57.63 | 59.12 |
|   SEPM (Fang et al., 2025) | 1.4923s | 51.66 | 42.56 | 51.15 | 86.93 | 70.87 | 86.39 | 58.65 | 58.50 | 60.25 |
|   **Ours** | 2.5953s | **53.29** | **44.02** | **53.91** | **87.63** | **76.15** | **87.59** | **61.17** | **61.56** | **62.46** |
| *Qwen3-VL-8B (Bai et al., 2025)* | | | | | | | | | | |
|   Zero-shot | 2.6964s | 52.78 | 43.35 | 52.50 | 85.57 | 65.89 | 85.72 | 65.94 | 63.89 | 66.16 |
|   Zero-shot-CoT (Kojima et al., 2022) | 14.5517s | 51.99 | 42.02 | 51.29 | 85.76 | 71.65 | 85.54 | 66.91 | 64.57 | 66.32 |
|   SparseVLM (Zhang et al., 2024c) | 6.390s | 52.60 | 42.36 | 52.33 | 87.31 | 73.09 | 87.02 | 66.91 | 64.57 | 66.32 |
|   SEPM (Fang et al., 2025) | 2.103s | 54.27 | 44.47 | 53.93 | 86.42 | 69.58 | 86.02 | 66.78 | 65.21 | 66.51 |
|   **Ours** | 3.0392s | **56.79** | **46.93** | **56.83** | **87.89** | **76.43** | **88.09** | **68.37** | **66.29** | **68.70** |

## 4.1. Datasets

We conducted experiments on three widely used multi-modal emotion datasets: MAFW (Liu et al., 2022), CH-SIMS (Yu et al., 2020) and MER23 (Lian et al., 2023). **MAFW** contains 10,045 video clips with 11 basic and 32 compound emotion labels; we follow the standard five-fold cross-validation protocol for comparisons. **CH-SIMS** is a Chinese dataset with 2,281 clips annotated by a continuous sentiment score in $[-1, 1]$ with modality-level annotations. **MER23** provides short video clips with aligned audio and text for 6 discrete emotion classification (neutral, anger, happiness, sadness, worry, surprise).

## 4.2. Experimental Setting

Following prior work (Liu et al., 2022), we adopt standard evaluation metrics for each dataset. For MAFW and MER23, we report Unweighted Average Recall (UAR), Weighted Average Recall (WAR), and macro-averaged F1-score. For CH-SIMS, we report 2-class accuracy (Acc-2), 3-class ac-

curacy (Acc-3), and macro-averaged F1-score. All experiments were conducted on a Linux workstation with a single NVIDIA A100 Tensor Core GPU, and our method was implemented in PyTorch. We evaluate WETR on representative MLLMs (InternVL3.5-4B/8B and Qwen3-VL-4B/8B). For all experiments, we perform inference with publicly released pre-trained checkpoints without extra training.

## 4.3. Comparison with State-of-the-Art Methods (RQ1)

We evaluate WETR on three MER benchmarks (MAFW, CH-SIMS, MER23) with four frozen MLLM backbones (InternVL3.5-4B/8B and Qwen2.5-4B/8B), and compare against Zero-shot, Zero-shot-CoT, and prior training-free baselines (as shown in Tab. 1). For Zero-shot-CoT, we append *"let's think step by step"* as a prompt enhancement. Across all datasets and backbones, WETR consistently improves MER without changing model parameters or architecture, demonstrating strong cross-model generalization. In particular, WETR brings clear gains over Zero-shot on both InternVL3.5-8B (+4.56% WAR on MAFW) and Qwen2.5-

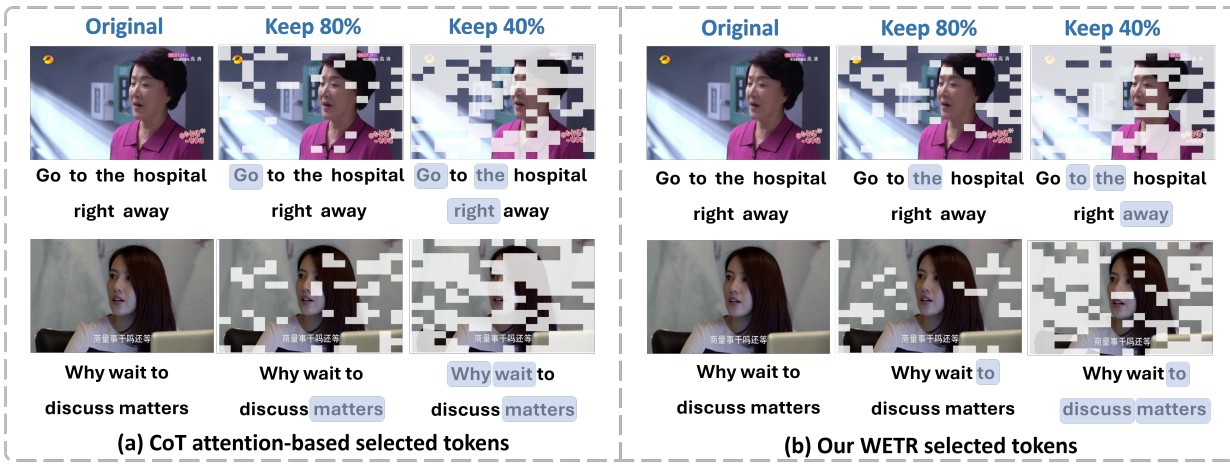

*Figure 3.* Visualization of selected tokens for different keep radios. The occluded part represents dropped tokens.

8B (+4.01% WAR on MAFW), and yields the largest improvement on MER23 (+7.79% WAR on InternVL3.5-8B). In terms of efficiency, WETR maintains comparable latency to Zero-shot, whereas Zero-shot-CoT is consistently much slower (about $4\times$-$7\times$ across backbones). Overall, WETR offers a favorable accuracy-efficiency trade-off and consistently improves MER across datasets and frozen MLLMs (answering **RQ1**).

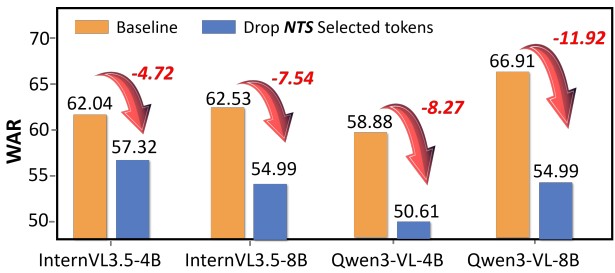

*Figure 4.* Effect of removing NTS-selected tokens across frozen MLLMs. Dropping the tokens selected by NTS causes a sharp WAR decrease on all backbones, indicating that NTS captures decision-critical emotion evidence.

### 4.4. Ablation Studies

**Comparison of Token Selection Strategies.** We study how the NTS scoring proxy affects evidence selection on MER23 with InternVL3.5-8B fully frozen. All variants keep the same token ratio and differ only in the scoring function (Tab. 2). Moving from Random selection to attention-based scoring consistently improves performance, indicating that NTS can identify a compact, informative evidence set rather than arbitrary tokens. Among these proxies, the proposed contrastive score is the most effective: it achieves the best WAR/UAR/F1 and improves UAR by +6.17% over Random (and +1.76% over using $Q_{\text{pos}}$ alone), showing that contrasting emotion-seeking and noise-seeking signals better suppresses $\varepsilon$-dominated redundancy and selects more emotion-relevant tokens (answering **RQ2**).

**Key Component Ablation Analysis.** We ablate WETR on MER23 to isolate the contributions of NTS and STR (Tab. 3). Both modules individually improve the frozen baseline, with NTS providing the larger boost and STR also yielding consistent gains. Combining both achieves the best WAR/UAR/F1, indicating that the two modules are complementary: NTS first selects a compact, emotion-relevant evidence set to suppress $\varepsilon$-dominated redundancy, and STR then reweights the retained tokens to further strengthen state-informative cues (supporting **RQ2-RQ3**).

*Table 2.* The effect of different proxy token-usage scoring strategies in NTS on the MER23 database. The best results are in **bold**.

| Selection strategies | WAR | UAR | F1 |
|---|---|---|---|
| Random | 64.48 | 57.07 | 64.56 |
| Query-related | 66.67 | 60.07 | 66.76 |
| $Q_{pos}$-related | 67.40 | 61.48 | 67.58 |
| Contrastive-query-related | **68.13** | **63.24** | **68.48** |

*Table 3.* The effect of different modules on model performance on the MER23 database. The best results are in **bold**.

| Baseline | NTS | STR | WAR | UAR | F1 |
|---|---|---|---|---|---|
| ✓ | | | 62.53 | 58.31 | 63.26 |
| ✓ | ✓ | | 67.64 | 61.68 | 67.84 |
| ✓ | | ✓ | 65.94 | 60.03 | 66.22 |
| ✓ | ✓ | ✓ | **68.13** | **63.24** | **68.48** |

**Visualization and Deletion Test of Selected Tokens.** Fig. 3 qualitatively shows what NTS removes under different keep ratios: it mainly drops $\varepsilon$-dominated visual background/redundant regions and weakly-informative text fragments, while preserving compact facial areas and decisive emotion triggers. To provide causal evidence, we further remove the NTS-selected tokens and keep all others unchanged. As shown in Fig. 4, this deletion causes a pronounced WAR degradation across all frozen backbones, indicating that the selected set is decision-critical. Together, these results confirm that NTS constructs a compact,

emotion-relevant evidence set by suppressing $\varepsilon$-dominated tokens (answering **RQ2**).

**Visualization of Inference-time Token Influence Redistribution.** Fig. 5 visualizes attention maps on MER23 to assess whether STR can redistribute token influence toward $z_e$-constraining cues. Compared with Base, Base+STR more often concentrates attention on facial regions, indicating that reweighting alone already provides a moderate sharpening effect. Meanwhile, without an explicit evidence filter, its attention may still be partially drawn to visually salient yet emotion-weak regions in some cases. By first filtering noisy tokens, NTS yields a cleaner evidence set and makes STR's reweighting more consistent. Consequently, WETR (NTS+STR) exhibits the most concentrated and stable attention on faces, supporting effective inference-time influence redistribution (answering **RQ3**).

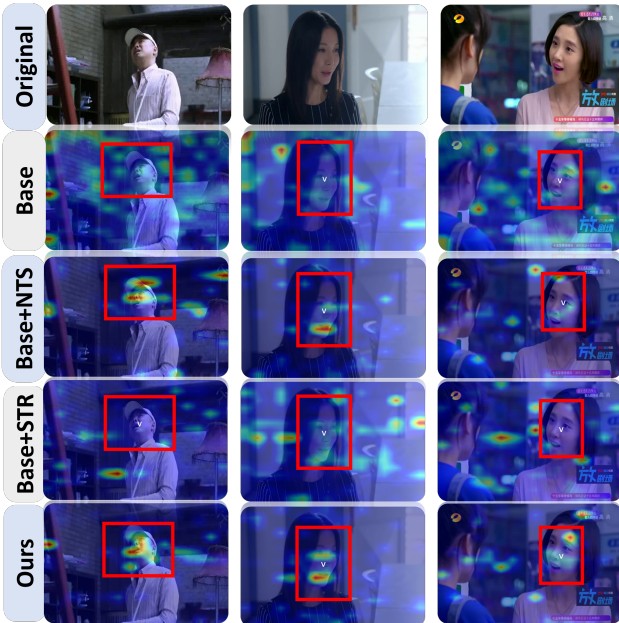

*Figure 5.* Attention heatmaps on MER23. We compare Base, Base+NTS, Base+STR, and WETR (Ours); red boxes mark facial regions. NTS sharpens attention by filtering evidence, while STR further concentrates attention on emotion-relevant facial cues; combining both yields the most stable and focused attention.

**Robustness Analysis under Noise Amplification.** To test robustness, we increase $\varepsilon$ to inject more emotion-irrelevant redundancy into both video and text with frozen InternVL3.5-8B. We report F1 and Emotion Attention Mass (EMass), which measures how much attention is allocated to a shared evidence set $\mathcal{K}$ under an emotion-seeking query versus a noise-seeking query (Fig. 6). As $\varepsilon$ increases, WETR degrades more slowly than the baseline and consistently preserves higher EMass, indicating that it better maintains $z_e$-constrained token usage under amplified nuisance. At the highest noise level ($\varepsilon{=}7$), WETR improves F1 over the baseline by 9.18%. Overall, WETR remains robust under

controlled nuisance amplification (answering **RQ4**).

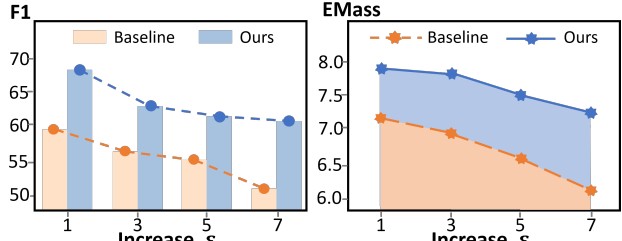

*Figure 6.* Robustness under noise amplification on the MER23 database. Left: F1 as $\varepsilon$ increases. Right: Emotion Attention Mass (EMass), measuring the attention mass captured by a shared evidence set under emotion-seeking vs noise-seeking queries.

**Parameter Sensitivity Analysis.** We study the sensitivity of WETR on MER23 (InternVL3.5-8B frozen) *w.r.t.* the trigger interval $n_r$, intervention strength $\delta$, and modality keep ratios $(p_v, p_t)$ (Tab. 4, Fig. 7). As shown in Tab. 4, both overly frequent and overly sparse refinements are suboptimal; a moderate interval performs best, and we set $n_r{=}9$. Similarly, $\delta$ is robust within a reasonable range, and $\delta{=}4$ gives the best overall results. Fig. 7 further shows that performance peaks at compact evidence sizes: keeping a small fraction of visual tokens ($p_v{=}0.03$) and a moderate fraction of text tokens ($p_t{=}0.2$) yields the best trade-off, while retaining too many tokens introduces $\varepsilon$-dominated redundancy and retaining too few removes useful cues. We use $n_r{=}9$, $\delta{=}4$, $p_v{=}0.03$, and $p_t{=}0.2$ for all experiments.

*Table 4.* The effect of different interval layers $n_r$ and intervention strength $\delta$ on the MER23 database. The best results are in **bold**.

| $n_r$ | WAR | UAR | F1 | $\delta$ | WAR | UAR | F1 |
|---|---|---|---|---|---|---|---|
| 2 | 66.67 | 61.75 | 67.11 | 1 | 67.40 | 61.60 | 67.50 |
| 3 | 67.40 | 59.81 | 67.17 | 2 | 67.15 | 61.98 | 67.41 |
| 6 | 66.91 | 60.95 | 67.10 | 3 | 67.64 | 62.98 | 68.04 |
| **9** | **68.13** | **63.24** | **68.48** | **4** | **68.13** | **63.24** | **68.48** |
| 18 | 66.42 | 61.69 | 66.87 | 5 | 68.07 | 62.64 | 68.33 |

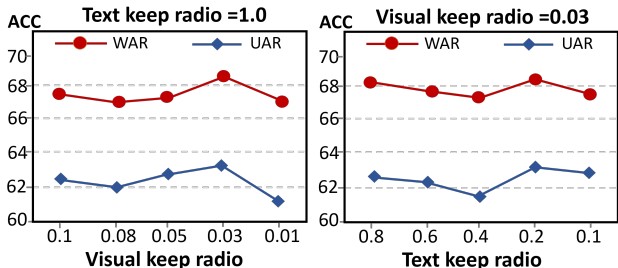

*Figure 7.* The effect of different modality keep ratios on the MER23 database.

**Error Propagation Analysis.** We study how irreversible token dropping affects inference on MER23 with InternVL3.5-8B fully frozen. Specifically, we inject controlled post-NTS selection noise with ratio $r$ and compare the frozen baseline, NTS-only, and full NTS+STR (Tab. 5).

*Table 5.* Error propagation analysis under controlled post-NTS noise injection on the MER23 database. Values are F1, with $\Delta$F1 from the clean setting shown in parentheses.

| Method | $r=0\%$ | $r=5\%$ | $r=10\%$ | $r=20\%$ | $r=30\%$ |
|---|---|---|---|---|---|
| Baseline | 60.57 (0.00) | 59.12 (-1.45) | 59.01 (-1.56) | 56.02 (-4.55) | 53.65 (-6.92) |
| NTS-only | 67.84 (0.00) | 67.62 (-0.22) | 67.23 (-0.61) | 66.24 (-1.60) | 65.14 (-2.70) |
| NTS+STR | **68.48** (0.00) | **68.26** (-0.18) | **68.04** (-0.44) | **67.46** (-1.02) | **66.72** (-1.76) |

As $r$ increases, performance degrades for all methods, but the full model degrades much more slowly than both the frozen baseline and NTS-only, indicating that STR effectively mitigates error accumulation after hard token selection. In particular, at $r=30\%$, the F1 drop of NTS+STR is only 1.76%, compared with 2.70% for NTS-only and 6.92% for the frozen baseline. We further examine the role of anchor protection in STR. As shown in Tab. 6, the full model consistently outperforms both NTS-only and the variant without anchor protection, suggesting that anchor protection further stabilizes inference after token filtering. Overall, these results show that although NTS is irreversible, its error propagation remains well controlled in WETR design.

*Table 6.* Effect of anchor protection on the MER23 database.

| Setting | WAR | UAR | F1 |
|---|---|---|---|
| NTS only | 67.64 | 61.68 | 67.84 |
| NTS + STR w/o anchor protection | 67.84 | 62.75 | 68.07 |
| NTS + STR (full) | **68.13** | **63.24** | **68.48** |

**Prompt Robustness of $q_{pos}$ and $q_{neg}$.** We evaluate three prompt variants for each query strategy on MER23 with InternVL3.5-8B fully frozen (Tab. 7). Using either $Q_{pos}$ or $Q_{neg}$ alone yields lower average performance and larger variance across prompt variants, indicating that single-query scoring is more sensitive to wording changes. By contrast, the proposed contrastive strategy achieves the best average F1 and the smallest standard deviation. Specifically, $Q_{pos} - Q_{neg}$ reaches 68.17% F1, outperforming $Q_{pos}$ only and $Q_{neg}$ only by 1.18% and 2.18%, respectively, while reducing the standard deviation to 0.40. These results suggest that the contrastive formulation provides a more stable estimate of emotion-relevant token utility.

*Table 7.* Performance and variance of different query scoring strategies under prompt variants on the MER23 database.

| Strategy | V1 | V2 | V3 | Mean$\pm$Std |
|---|---|---|---|---|
| $Q_{pos}$ only | 67.58 | 67.15 | 66.24 | 66.99$\pm$0.68 |
| $Q_{neg}$ only | 66.76 | 66.72 | 64.49 | 65.99$\pm$1.30 |
| $Q_{pos} - Q_{neg}$ | **68.48** | **67.71** | **68.32** | **68.17$\pm$0.40** |

**Token Decision Contribution Shift Analysis.** We study how STR affects the decision contribution of retained tokens on MER23 with InternVL3.5-8B fully frozen. Specifically, we group retained tokens by their reweighting factor $\gamma_i$ into three categories: downweighted (DW), neutral (NEU), and upweighted (UW), and compare their average contribution before and after STR, where contribution is mea-

sured by the drop in the target prediction score when the token is removed once (Tab. 8). Tokens with larger positive reweighting factors show a clear contribution increase after STR, whereas downweighted tokens exhibit a substantial decrease, indicating that STR does not simply rescale all retained tokens uniformly. In particular, the contribution of UW tokens increases by 0.0225, while that of DW tokens decreases by 0.0196; by contrast, NEU tokens remain nearly unchanged. These results suggest that STR selectively redistributes token influence by strengthening more state-informative evidence and suppressing weaker or more replaceable cues, supporting its role as an inference-time contribution modulator (answering **RQ3**).

*Table 8.* Token decision contribution shift before and after STR on the MER23 database.

| Group | Avg.$\gamma_i$ | Before STR | After STR | Change |
|---|---|---|---|---|
| DW | 0.000 | 0.0301 | 0.0105 | -0.0196 |
| NEU | 1.000 | 0.0131 | 0.0139 | +0.0008 |
| UW | 2.737 | 0.0381 | 0.0606 | **+0.0225** |

## 5. Conclusion

We propose **WETR**, a training-free, plug-and-play inference-time regulator for multimodal emotion recognition with frozen MLLMs. Motivated by a world-model perspective, WETR improves reliability by making token usage explicit and refining it during inference, reducing the dominance of redundant or irrelevant context while strengthening decision-relevant emotional evidence that better constrains the latent emotion state $z_e$. WETR performs sparse interventions at a few trigger layers via a lightweight selection-reweighting loop: **NTS** acts as an observation filter to admit a compact set of emotion-relevant tokens, and **STR** serves as a likelihood modulator to redistribute token influence using state-gated cross-modal predictability with anchor protection. Extensive experiments on MAFW, CH-SIMS, and MER23 across multiple frozen backbones demonstrate consistent improvements, together with more interpretable token-level evidence usage.

## Acknowledgements

This work was supported by the National Natural Science Foundation of China grant (62076227), Natural Science Foundation of Hubei Province grant (2023AFB572) and Hubei Key Laboratory of Intelligent Geo-Information Processing (KLIGIP-2022-B10).

## Impact Statement

This paper presents work whose goal is to advance the field of Machine Learning. There are many potential societal consequences of our work, none which we feel must be specifically highlighted here.

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
