# OpenReview forum: "World-Model Inspired Emotion-aware Token Refinement for Training-Free Multimodal Emotion Recognition"
_ICML.cc/2026/Conference — ICML 2026 spotlight_

### Official Review · Reviewer_EsuZ · 2026-03-11

**Soundness:** 2
**Presentation:** 2
**Significance:** 2
**Originality:** 2
**Overall Recommendation:** 3
**Confidence:** 3

**Summary:**

In this paper, a training free, plug and play inference time regulator named WETR is proposed to improve the performance of the multimodal large language model with frozen parameters in multimodal emotion recognition tasks. From the perspective of a novel "world model", this method regards emotion as a potential state that needs to be inferred from noisy observations. Its core is to optimize the use of token iteratively in the sparse trigger layer through two mechanisms: the noise suppression token selection module is used to screen high information emotional related evidence, and the state enhancement token reweighting module is used to adjust the influence of reserved tokens. Experiments on several benchmark datasets show that this method can consistently improve the accuracy, stability and token level interpretability of the model.

**Compliance With Llm Reviewing Policy:**

Affirmed.

**Key Questions For Authors:**

1. How to more formally establish the mathematical connection between the "world model" formula (formula 1) and the nts/str operation (especially the comparison score of formula 4 and the residual/overlap score of formula 10)? Is it possible to derive a computable surrogate objective function for the uncertainty of potential state z_e under the condition of frozen model, so as to directly optimize it?
2. What are the specific prompts of q_pos and q_neg? Does their effectiveness depend heavily on specific prompt wording? How to ensure that q_neg query can really attract "noise driven" attention, rather than accidentally capture some context related to emotional implication? Has the robustness analysis of prompt words been carried out?
3. Formula 8 interprets \gamma_i as the modulation weight of token likelihood term log P (x_i|z_e). In the forward propagation of frozen transformer, can the \gamma_i realized by simply scaling the hidden state h_i be equivalently interpreted as this modulation in the sense of probability? Is there any experimental evidence (for example, before and after the weight adjustment, token indicates the change of contribution to the final decision) to support this "modulation" explanation, rather than merely "enhancing/weakening some characteristics"?
4. The wetr method does not seem to be specific to emotion recognition, and its core idea (suppressing noise and strengthening decision-making evidence) can be applied to other tasks that need to extract key information from redundant multimodal inputs. Has the author tested the performance of this method on other tasks (such as multimodal implication, fine-grained visual question answering)? In addition, how effective is this method in dealing with samples with serious missing modes or serious signal conflicts between modes?
5. Although the paper shows that wetr is much faster than zero shot cot, it still has computational overhead compared with native zero shot. Can we quantitatively analyze the proportion of additional flops or time brought by the comparison query calculation in NTS and the cross modal predictability calculation in str? In extreme resource constrained scenarios, is there a lighter simplified version?

**Limitations:**

The limitations of the method proposed in this paper have not been demonstrated. It is suggested to add a separate chapter to discuss this.

**Strengths And Weaknesses:**

Strengths:
1. The MER problem is reformulated from the perspective of "world model" and potential state inference, which provides powerful theoretical inspiration for understanding and improving the reasoning process of frozen model.
2. The proposed WETR framework does not need training and is plug and play. The NTS and STR modules are complementary in design, starting from the two ends of evidence access and influence modulation respectively, with clear logic.

Weaknesses:
1. This paper compares emotion recognition to inferring potential world states from noisy observations, which is an attractive metaphor. However, the theoretical connection between this formalization and the proposed specific method (nts/str) is not close and rigorous enough. The mathematical correlation between the difference of attention and information gain of comparative query and the strict probability interpretation of "likelihood modulation" in str used by nts are heuristic and indirect, which are more like an efficient heuristic method inspired by this perspective than a strictly derived inference algorithm. The potential of the world model framework has not been fully tapped to provide a more solid theoretical guarantee.
2. The specific structure text of the comparison query (q_pos and q_neg) in nts is not clearly given in the text, which is very important for understanding how to accurately distinguish emotion related and unrelated signals, and also affects the recurrence.
In str module, although it is intuitive to distinguish between "replaceable overlap" and "complementary residual evidence" through cross modal predictability, there is no theoretical or empirical evidence to show that this can reliably correspond to the size of "providing incremental constraints on potential state z_e". The design of the coupling mode (equation 10) between the normalized error e_i and the NTS fractional gating g_i is slightly vague, and the sensitivity of its super parameters (such as the scaling of the sigmoid function) is not discussed.
The protection mechanism for "anchor points" is an important stability measure, but the criteria for selecting anchor points (based on NTS score only) and the setting of the lower limit of weight \gamma_anc are not fully demonstrated.
3. Although the comparison baseline includes mainstream free training methods, it is not enough compared with the most advanced methods specially designed for mer and may be combined with light-weight fine-tuning, and it fails to fully locate the position of WETR in the progress of the whole field.
The experiments are carried out on models with less than 10 billion parameters, and the effectiveness and efficiency characteristics on larger scale (such as 70billion parameters) mllm are not clear.
The simulation of "noise" (ε amplification experiment) may be too simplified, and the redundancy and context independence of real scenes are more complex.
4. Some expositions are suspected of slight circular argumentation or exaggeration. For example, the performance improvement is directly attributed to "better constraint on the potential emotional state z_e", but this is an assumption that has not been independently verified. Ascension may result from a more general focus mechanism. The claim of "more interpretable token level evidence use" in the conclusion is mainly supported by attention visualization and lacks more rigorous attribution analysis.

---

> ### Author Rebuttal · Authors · 2026-03-31
>
> Dear Reviewer EsuZ:
>
> Thanks for the insightful comments. Our responses follow.
> # Q1:Connection between the world-model view and NTS/STR.
> We would like to clarify that WETR is world-model-inspired in a latent-state sense, not a simple application of world-model formation. This view provides the design principle; NTS and STR are practical proxy implementations. We adopt it because emotion recognition here is latent-state inference: useful cues are sparse and noisy, so a state-centric strategy is more suitable than direct feature matching or heuristic token filtering.
>
> Eq.(1) formulates emotion recognition as inferring a latent emotional state $z_e$ from noisy observations. Because emotional signals are subtle and masked by context, inference under frozen MLLMs should select and weight cues better constraining the latent state, motivating NTS and STR. Because $z_e$ is unobserved and not explicitly parameterized, formal latent-state inference is infeasible. Thus, Eq.(4) and Eq.(10) are designed as proxy mechanisms that approximate this latent-state inference objective while remaining implementable at inference time. Here, $Q_{pos}-Q_{neg}$ in NTS is not information gain, but a contrastive proxy that suppresses shared/context-driven attention and highlights relevant evidence; the likelihood modulation view in STR is intuitive, not a strict probabilistic formulation.
>
> A direct uncertainty objective over $z_e$ would be interesting, but posterior uncertainty and conditional entropy are inaccessible under frozen MLLMs. Optimizing them would require latent-state estimation, posterior approximation, or test-time optimization, beyond the training-free, inference-time token control setting. We will clarify this in revision.
> # Q2:Prompt robustness of $q_{pos}$ and $q_{neg}$.
> We add the exact prompts:
> - $q_{pos}$: Please focus on decision-critical emotion cues/triggers (facial actions, body language, negation, intensity, contrast, sentiment).
> - $q_{neg}$: Please focus on objective facts and scene description (entities, actions, time, location, background), not emotion judgments.
>
> The method does not assume that $q_{pos}$ captures only emotion or that $q_{neg}$ captures only noise. Their attended regions partially overlap, which is why we use $Q_{pos}-Q_{neg}$: $q_{neg}$ serves as a generic non-emotional baseline that subtracts shared attention.
> |Strategy|V1|V2|V3|Mean±Std|
> |-|-|-|-|-|
> |$Q_{pos}$ only|67.58|67.15|66.24|66.99±0.68|
> |$Q_{neg}$ only|66.76|66.72|64.49|65.99±1.30|
> |$Q_{pos}-Q_{neg}$|68.48|67.71|68.32|**68.17±0.40**|
>
> # Q3:Eq.(8), Eq.(10), and evidence for STR.
> Eq.(8) should not be read as claiming that scaling $h_i$ by $\gamma_i$ in a frozen Transformer is a strict probabilistic reparameterization of $\log p(x_i|z_e)$. Our intended claim is more limited: STR changes the relative contribution of retained tokens to downstream aggregation and final prediction. To test this directly, we add a Token Decision Contribution Shift analysis:
> |Group|Avg.$\gamma_i$|Before STR|After STR|Change|
> |-|-|-|-|-|
> |Top-downweighted tokens|0.000|0.0301|0.0105|-0.0196|
> |Neutral tokens|1.000| 0.0131| 0.0139 |+0.0008|
> |Top-upweighted tokens|2.737|0.0381|0.0606|**+0.0225**|
>
> We also add sensitivity analysis for Eq.(10) and anchor protection:
> |Sigmoid Scale|F1|
> |-|-|
> |0.5|68.27|
> |1.0|68.48|
> |2.0|**68.68**|
> |4.0|68.46|
>
> |Anchor Ratio $p_{anc}$|$γ_{anc}$|F1|
> |-|-|-|
> |1%|1.00|67.30|
> |5%|1.00|**68.48**|
> |5%|1.20|67.95|
> |10%|1.20|67.88|
> |20%|1.20|67.86|
>
> |Anchor Selection|F1|
> |-|-|
> |No anchors|68.07|
> |Random anchors|67.85|
> |NTS-score anchors (ours)|**68.48**|
> # Q4:Task scope, robustness, efficiency, and larger backbones.
> WETR's token-control principle is not limited to MER and may generalize to other multimodal tasks with sparse, decision-critical cues. We focus on MER as the motivating case, using emotion-specific queries and evidence criteria, and present it as a MER-centered instance of a broader inference-time refinement idea.
>
> We add the following missing or conflicting modalities analyses:
> |Setting|Baseline|Ours|
> |-|-|-|
> |Full input|67.64|68.13|
> |Missing-text|61.07 (-6.57)|63.06 (**-5.07**)|
> |Missing-visual|39.77 (-27.87)|46.07 (**-22.06**)|
>
> |Subset|Baseline|Ours|
> |-|-|-|
> |CH-SIMS conflict subset|75.39|78.34 (**+2.95**)|
>
> We further quantify module-level overhead and larger-scale backbones:
> |Method|Time(s)|GPU(M)|F1|
> |-|-|-|-|
> |Zero-shot|1.0561|16862|60.57|
> |NTS-only|1.2522|16874|67.84|
> |STR-only|1.2563|16874|66.22|
> |Full WETR|1.4660|16874|**68.48**|
>
> |Backbone|Zero-shot F1|WETR F1|Zero-shot Time\|GPU|WETR Time\|GPU|
> |-|-|-|-|-|
> |InternVL3_5-14B|63.74|67.15(**+3.41**)|3.1052s\|30292M|3.8020s\|30314M|
> |InternVL3_5-30B-A3B|67.55|69.35(**+1.80**)|9.5627s\|65768M|10.1348s\|65734M|
>
> Due to compute limits, we cannot provide a reliable 70B result, but we do report on larger 14B and 30B backbones.
> # Q5: Inappropriate expressions.
> We will revise the inappropriate expressions in the final version. Thanks for your suggestions.

---

> > ### Author Rebuttal · Reviewer_EsuZ · 2026-04-04
> >
> > Thank you for the author's response. But I still feel that this paper is like a gimmick of a world model,  lacking theoretical basis and many unclear areas.
> >
> > 1) What does the phrase 'emotion recognition can be inherently a problem of late state inference rather than direct pattern matching' mean in the introduction (L088-089)? Is there any theoretical basis or is this common sense?
> >
> > 2)The introduction claims that 'the world model assumption that robust emotion recognition requires prioritizing evidence that constrains the latent state”. But which work has this assumption about emotion recognition? Please provide the source and theoretical proof? Is this assumption reasonable? Or is it just a gimmick to put on the world model?
> >
> > 3) As is well known, training-free methods generally rely heavily on prompts/instructions and may not be as stable, but the authors did not provide any code for inspection? No theoretical evidence has been provided to demonstrate the effectiveness of the method proposed in this paper. So I think the current experimental results are not convincing. Even if there is not enough space in the main text, the theoretical proof can be placed in the appendix. The current methods and experiments are entirely empirical.
> >
> > 4) The proposed  is actually universal, modular combination, incremental, and does not indicate that it is designed for emotion recognition  (as Q4), without a clear and intuitive understanding. So I always believe that this paper is more of a hype about the concept of the world model in MER. As for whether it really works, it cannot be judged, and the author's motivation is unclear. I think the paper needs a major revision.
> >
> > Additionally, we hope that other reviewers will carefully consider the credibility of this paper.

---

> > > ### Author Response · Authors · 2026-04-05
> > >
> > > Dear Reviewer EsuZ:
> > >
> > > Thank you for the opportunity to further clarify our work. We respond below with additional support.
> > >
> > > # Q1: On latent-state inference in emotion recognition
> > > The view of emotion as latent-state inference has been increasingly studied. Prior works [1,2] treat emotion as a latent, unobservable internal state inferred from multiple cues, including situations, behaviors, and expressions. Recent work further shows that emotion inference depends on integrating facial and contextual cues, rather than being read off from a single surface pattern [3]. Thus, for MER, viewing emotion as a latent state inferred from noisy multimodal observations is a literature-grounded and well-supported. We will add this discussion.
> > > # Q2: On prioritizing latent-state-constraining evidence
> > > We emphasize that this is a literature-grounded motivation, not a formal theorem. Prior work treats emotion as a latent internal state inferred from situations, behaviors, and expressions [1,2]. Recent studies also show that emotion recognition is often distracted by background bias and redundant non-emotional cues, which divert models from decision-relevant evidence [3,6]. Under this view, not all evidence should contribute equally; robust inference should prioritize cues that more strongly reduce uncertainty about the latent emotional state. This is why a world-model-inspired perspective is relevant: world-model frameworks are designed for latent-state reasoning under partial or noisy observations [4,5]. Recent emotion-centered MLLM works move similarly by sharpening emotion-related triggers or evidence spans instead of treating all tokens equally [6]. However, such principles have not yet been systematically introduced into this setting. Our method instantiates this view as a task-specific token refinement mechanism for MER, and we will clarify this more explicitly in the paper.
> > > # Q3: On stability, reproducibility, and theoretical grounding
> > > On stability. Indeed, many training-free approaches, e.g. Visual Programming [7] and CoT [8], rely heavily on prompts and can therefore be sensitive to prompt variations. In contrast, our method takes a different route: WETR operates independently of prompt design, performing token-level refinement inside frozen MLLMs during inference. This makes it less dependent on prompt variation than purely prompt-based  methods. Our results also support this distinction: WETR consistently improves performance across multiple backbones and datasets without prompt tuning. We will clarify this more explicitly in the revision.
> > >
> > > On code availability. We appreciate the concern about reproducibility. We will release the code upon acceptance and provide more implementation details in the appendix to facilitate inspection and reproduction.
> > >
> > > On theoretical justification. Although our method is mainly validated empirically, it is grounded in the latent-state inference framework widely used in world-model research [4,5,9].  Specifically, we treat emotion as a latent state, and token refinement as selecting and weighting evidence that constrains this state. Thus, the method is not purely heuristic, but conceptually grounded in established world-model reasoning.
> > >
> > > Moreover, we support this foundation with consistent empirical gains across datasets and backbones (avg. **+3.34% F1**), together with ablations (Table 3; Fig. 6–7). We will further clarify this motivation in the revision.
> > > # Q4: On task specificity and motivation for MER
> > > We emphasize two points: (1) Our motivation comes from a concrete MER-specific gap: emotional cues are often sparse, subtle, and easily overwhelmed by redundant multimodal context. Accordingly, our solution is task-driven and empirically effective. (2)  We formulate MER from a world-model-inspired view by explicitly modeling noise from emotion-irrelevant background. As a result, our method is directly derived from the MER task.  While it may also extend to related areas such as micro-expression recognition and sarcasm understanding, its current motivation and design are centered on MER. We will explore these extensions in future work.
> > >
> > > [1] Ong, Zaki, Goodman. Affective Cognition: Exploring Lay Theories of Emotion. Cognition, 2015
> > >
> > > [2] Ong, Zaki, Goodman. Computational Models of Emotion Inference in Theory of Mind: A Review and Roadmap. TopiCS, 2019
> > >
> > > [3] Goel et al. Face and context integration in emotion inference is limited and variable across categories and individuals. Nat. Commun., 2024
> > >
> > > [4] Ha & Schmidhuber, World Models, NeurIPS 2018
> > >
> > > [5] Hafner et al. Dreamer, ICLR 2020
> > >
> > > [6] Fang et al. Catch Your Emotion: Sharpening Emotion Perception in Multimodal Large Language Models. ICML, 2025
> > >
> > > [7] Gupta et al. Visual programming: Compositional visual reasoning without training. CVPR, 2023
> > >
> > > [8] Wei et al. Chain-of-thought prompting elicits reasoning in large language models. Adv. Neural Inf. Process. Syst., 2022
> > >
> > > [9] Hafner et al. Learning Latent Dynamics for Planning from Pixels. ICML, 2019

---

### Official Review · Reviewer_kVG7 · 2026-03-12

**Soundness:** 4
**Presentation:** 2
**Significance:** 3
**Originality:** 3
**Overall Recommendation:** 5
**Confidence:** 5

**Summary:**

This paper studies how to make frozen multimodal LLMs more stable for multimodal emotion recognition by adjusting how tokens are used during inference. The authors propose a training-free framework called WETR, which mainly includes two components: NTS for selecting a compact set of emotion-related tokens, and STR for further reweighting these tokens so that more informative emotional cues have stronger influence during reasoning. The experiments on several MER benchmarks show consistent improvements over standard zero-shot and zero-shot-CoT baselines. Overall, the idea of controlling token usage at inference time is interesting and the NTS/STR design is quite detailed. However, the paper heavily frames the method using a “world-model” perspective, while the actual technique is closer to token filtering and reweighting, which makes the conceptual framing somewhat disconnected from the implemented method and also hurts the readability of the paper.

**Compliance With Llm Reviewing Policy:**

Affirmed.

**Final Justification:**

Good paper, and I will accept this paper.

**Key Questions For Authors:**

**Key Questions For Authors**

1. The paper heavily frames the method using a world-model perspective. Could the authors clarify why a world-model formulation is necessary for this problem?

2. NTS relies on two additional queries to compute attention-based scores. How sensitive is the method to the specific design of these queries?

3. The framework is described as plug-and-play for frozen MLLMs. Can it realistically be applied to closed models (e.g., GPT-style APIs) where internal attention and hidden states are not accessible?

4. STR introduces a relatively complex mechanism for distinguishing overlap and residual evidence. How critical is this specific design compared with simpler reweighting strategies?

5. The introduction claims that model decisions may not be grounded in informative emotional evidence. Do the authors have empirical evidence supporting this statement?

6. Have the authors compared their method with stronger reasoning strategies (e.g., self-consistency CoT or other multi-step reasoning approaches)?

**Limitations:**

See the above weaknesses.

**Strengths And Weaknesses:**

**Strengths**

1. The overall method design is quite careful and detailed. The paper decomposes the framework into NTS and STR, and each component has a relatively clear functional role in the pipeline.
2. The design of NTS and STR is interesting. In particular, the contrastive query idea for selecting emotion-related tokens and the later reweighting mechanism for distinguishing overlapping versus residual evidence are thoughtful engineering choices.
3. Although the experimental setup is not extremely comprehensive, the results are generally consistent with the claims of the paper and the ablation studies do provide some support for the proposed design.

**Weaknesses**

1. In the abstract, the paper claims that inference-time methods such as chain-of-thought mainly rely on heuristic prompting and do not focus on internal emotion-related tokens. This statement is somewhat inaccurate, since there are already several inference-time approaches that manipulate hidden states or token usage internally rather than relying purely on prompts.
2. In the introduction, the statement that “even if MLLMs generate a reasonable emotional response, the decision may not be grounded in informative emotional evidence” is quite strong, but the paper does not provide clear empirical evidence to support this claim.
3. The connection to the “world model” concept is weak. Although the paper describes emotion as a latent state, emotion recognition in this setting is still essentially modeling conditional probabilities over inputs. It is not clear why the world-model perspective is necessary here.
4. In practice, the method looks much closer to a token filtering / token skipping strategy that tries to keep emotion-relevant tokens. This seems largely unrelated to world-model reasoning, and the paper would likely be clearer if the world-model narrative were either better justified or removed.
5. The paper claims the framework is plug-and-play, but the method still modifies the inference process of the model. It is therefore unclear how easily it can be applied to closed models such as GPT-5 or Claude-4, where internal attention and hidden states are not accessible.
6. In the proxy scoring step, two additional queries are introduced and attention scores are used to select tokens. This raises the question of how sensitive the method is to the specific queries. If the queries change, will the selected tokens fluctuate significantly?
7. The STR component is quite a fine-grained engineering design involving overlap and residual evidence, but the internal analysis of STR and NTS is still somewhat limited. More detailed investigation would help understand why the method works.
8. The readability of the paper is rather poor. The core idea is actually quite straightforward, but the theoretical framing and technical descriptions make it unnecessarily difficult to follow. Several figures (e.g., Fig. 2) could also be improved.
9. Finally, it would be interesting to know whether stronger reasoning strategies, such as self-consistency CoT or more task-specific reasoning frameworks (e.g., methods similar to SarcasmCue), could achieve comparable results. This is not required for rebuttal, but it is a natural question when reading the paper.

---

> ### Author Rebuttal · Authors · 2026-03-31
>
> Dear Reviewer kVG7,
>
> Thank you for your thoughtful and constructive feedback. We address your concerns point by point below.
>
> # Q1：On the role of the “world-model” view.
> Thank you for raising this. We would emphasize that our method is world-model-inspired in a latent-state sense, rather than aiming to instantiate a classical world model with explicit entities, dynamics, or temporal simulation. Specifically, we adopt a latent-state inference perspective, where multimodal emotion recognition is viewed as estimating an unobserved emotional state from noisy multimodal observations. Under frozen MLLMs, this perspective provides a state-centric criterion for token refinement: a token is useful not merely because it is salient, but because it helps constrain the latent emotional state. This view directly motivates our design: the NTS determines which observations should be admitted into inference, while STR adjusts the contribution of retained cues when estimating the latent state. In fact, the world-model-inspired perspective mainly provides a principled rationale for organizing these operations, rather than treating them as heuristic filtering. Overall, the world-model perspective is not an additional narrative layer, but a conceptual foundation that guides the design of our novel token refinement mechanism. To avoid potential misunderstanding, we will further revise the wording to clarify that we adopt a latent-state world-model perspective, rather than simply implementing a world-model in the task. We will also revise the abstract and introduction to more clearly position our method relative to prior inference-time token refinement and test-time adaptation approaches.
> # Q2：On "plug-and-play".
> Thank you for pointing this out. We will clarify the applicability boundary in the revision. In our intended usage, “plug-and-play” means no training, no parameter updates, and no architecture redesign for frozen open-weight MLLMs with accessible hidden states and attention. It does not apply to closed-source models via APIs. Regarding the benefits to other models, we tried to apply on more powerful open source models and can observe consistent gains as below:
>
> |Backbone|Zero-shot F1|WETR F1|
> |-|-|-|
> |InternVL3_5-14B|63.74|**67.15 (+3.41)**|
> |InternVL3_5-30B-A3B|67.55|**69.35 (+1.80)**|
>
> # Q3：On $q_{pos}$ / $q_{neg}$ design.
> We add the exact prompts in the revision:
> - $q_{pos}$: Please focus on decision-critical emotion cues/triggers (facial actions, body language, negation, intensity, contrast, sentiment).
> - $q_{neg}$: Please focus on objective facts and scene description (entities, actions, time, location, background), not emotion judgments.
>
> We do not assume $q_{pos}$ captures only emotion and $q_{neg}$ captures only noise.  In practice, the two prompts partly overlap in attended regions and tokens. We therefore use their difference, where $q_{neg}$ provides a generic baseline that cancels shared attention and leaves more decision-relevant emotional evidence. We also tested alternative prompt wordings:
>
> |Strategy|V1|V2|V3|Mean±Std|
> |-|-|-|-|-|
> |$Q_{pos}$ only|67.58|67.15|66.24|66.99±0.68|
> |$Q_{neg}$ only|66.76|66.72|64.49|65.99±1.30|
> |$Q_{pos}-Q_{neg}$|68.48|67.71|68.32|**68.17±0.40**|
>
> # Q4: On whether STR’s specific design is necessary.
> Thank you for this question.  To address this, we compared full STR with a simpler score-only reweighting under fixed NTS:
>
> |Method|F1|
> |-|-|
> |NTS only|67.84|
> |NTS + Simple Reweight (score-only)|67.90|
> |NTS + Full STR (Our)|**68.48**|
>
> # Q5:  On the grounding claim.
> Thank you for this point. We will revise the introduction to make the claim more precise. Our point is only that MER may sometimes produce plausible predictions without stable focus on decision-relevant emotional evidence. Existing results already provide partial evidence: the ablation test shows that removing NTS-selected tokens causes substantial degradation, and the attention analysis shows that the baseline is more distractible and degrades faster under noise. We further add a grounding–stability analysis: we use attention mass as the grounding score, split test samples by the median into high-/low-grounding subsets, and compare perturbation stability:
>
> |Subset|Consistency Rate|F1 Drop (Clean vs. Perturbed)|
> |-|-|-|
> |Low-grounding|86.82|3.62|
> |High-grounding|**89.80**|**0.85**|
>
>
> # Q6: Comparison with other reasoning baselines.
> Thank you for this suggestion. We added self-consistency CoT for comparison. The results show that stronger reasoning alone improves over vanilla CoT, but WETR remains both more effective and much more efficient:
>
> |Method|F1|Time (s)|
> |-|-|-|
> |Zero-shot| 60.57| 1.0561|
> |Zero-shot-CoT| 58.19| 7.1399|
> |Self-consistency CoT| 64.92| 40.0729|
> |WETR|**68.48**|1.4660|
>
> # Q7: On wording and figures.
> Thank you for this valuable feedback. We will revise the wording and figures in the final version for better clarity and readability.

---

### Official Review · Reviewer_UMAX · 2026-03-12

**Soundness:** 2
**Presentation:** 3
**Significance:** 3
**Originality:** 2
**Overall Recommendation:** 4
**Confidence:** 2

**Summary:**

In this paper, authors identify a diluted emotional info issue from multimodal data (video and text, no audio). They propose to enhance the MLLM emotion identification by adding steps at the inference-time. To do so, they propose a regulation to focus on correct emotional cues by suppressing the redundancy and weighting more the emotion cues. This is a feature selection process on trained multimodal models at inference time: a two-step process comprising of a intra-modal filter (selection dropping features) and a cross-mmodal modulator (weighting features). Both steps are intuitive and follow an detailed logic. Authors evaluate their approach on 3 datasets, two using categorical values and one using continuous values.

**Compliance With Llm Reviewing Policy:**

Affirmed.

**Final Justification:**

Everything is already stated above. I updated my score from reject to weak accept because my frst assessment (reject) was harsh compared to my explanation so please consider it a move from weak reject to weak accept. I already said all of this in the review and the rebuttal assessment. I think this is a good paper with a good hypothesis, but a limitation of the error dependency.

**Key Questions For Authors:**

- what about using this to further update the models? did you try to do so?
- did you quantify the error propagation rate?

**Limitations:**

No. Various societal impacts could arise, but no specific ones. However, more limitations discussions on error traces should be in the paper.

**Strengths And Weaknesses:**

# strengths
- interesting hypothesis on detailed emotional cues and subtle ones
- the approach does not need training, just a steering two-step process
- the synergy between the two selection steps, the filter and the modulator, is interesting
- Figure 5 is impressive, leading to indicate potential explanability usage for the propose approach

# weaknesses
- we may argue that such feature selection could be trained, or at least aligned, to better yield an automatic selection
- the world model expression seems a stretch for this paper, why not just say it is test-time feature selection? world-model is learning from modelling elements in the world, not only inference
- the STR phase depends on the NTS phase. It is a shame that such elements dropped from the NTS would never be able to come back due to a STR reweightin process. Hence, the error propagation is fully there. This would complement table 3

---

> ### Author Rebuttal · Authors · 2026-03-31
>
> Dear Reviewer UMAX:
>
> We sincerely appreciate your time and effort in reviewing our paper. We have provided further clarification on each of the issues you raised.
> # Q1: Why not further train the selector or update the model?
> We appreciate this suggestion. However, we clarify that the primary goal of this work is to investigate whether effective token refinement can be achieved under a stricter and practically relevant training-free setting, rather than training or fine-tuning an additional selection network. Introducing a trainable selector would fundamentally shift the problem from training-free adaptation to trainable adaptation, which represents a different research direction and is beyond the scope of this work.
>
> The motivation for a training-free design is primarily practical. Many real-world deployments rely on frozen large-scale MLLMs, where additional training could be either impractical or undesirable due to computational cost and limited access to model parameters. Moreover, training-free approaches enable direct applicability across different MLLMs without additional optimization, thereby improving transferability and scalability. Furthermore, our approach goes beyond prompt engineering by directly regulating internal token usage during inference, rather than operating only at the input level. This provides a more principled and fine-grained mechanism for controlling model reasoning under frozen parameters, which we believe could contribute good insights to the community.
> # Q2: Why use the “world-model-inspired” framing?
> We clarify that the WETR is inspired by the latent  state inference view from the world model framework rather than generic feature filtering. We use the term world-model-inspired because, in the training-free setting, MER is better viewed as estimating an unobserved emotional state from noisy multimodal observations. Emotional cues are inherently sparse, indirect, and embedded in noisy context, making latent-state inference more suitable than direct feature matching when model parameters remain frozen.
>
> In this sense, our use of the world-model perspective is conceptual rather than architectural. Under this formulation, tokens act as observations, and token refinement corresponds to controlling how evidence is admitted and weighted for latent-state inference. This provides a principled, state-centric rationale for WETR, beyond prompt engineering, heuristic token filtering, or direct feature aggregation.
> More specifically, this view explains the roles of the two modules: NTS determines which observations should enter inference, while STR adjusts how strongly retained evidence contributes to estimating the latent state. Therefore, “world-model-inspired” describes the organizing principle of our training-free token refinement, rather than claiming a full classical world model with explicit entities, relations, or temporal dynamics. This perspective offers conceptual grounding beyond conventional test-time feature selection while remaining compatible with frozen MLLMs. We will revise the manuscript to make this scope and motivation clearer.
> # Q3:  On NTS→STR irreversibility and error propagation.
> This is a good question. Indeed, NTS is implemented by a hard selection step, so tokens dropped by NTS cannot be reintroduced by STR within the same inference pass. Regarding this, our goal is not to eliminate this irreversibility, but to keep it controllable under a training-free setting. In particular, the current design already includes several mitigation mechanisms: (1) modality-wise top-p retention instead of aggressive global pruning; (2) always preserving instruction tokens to maintain prompt stability; and (3) anchor protection in STR for high-confidence evidence.
>
> To directly quantify error propagation, we added an error propagation analysis: by injecting controlled post-NTS selection noise and comparing against NTS-only and matched random-corruption baseline, we find that propagation is present but substantially mitigated in full NTS+STR.
>
> | Noise Injection Ratio $r$ |Baseline F1 ( ΔF1 vs. clean)|NTS-only F1 ( ΔF1 vs. clean)|our NTS+STR F1 ( ΔF1 vs. clean)|
> |-|-|-|-|
> |0%|60.57 (0.00)|67.84 (0.00)|**68.48 (0.00)**|
> |5%|59.12 (-1.45)|67.62 (-0.22)|**68.26 (-0.18)**|
> |10%|59.01 (-1.56)|67.23 (-0.61)|**68.04 (-0.44)**|
> |20%|56.02 (-4.55)|66.24 (-1.60)|**67.46 (-1.02)**|
> |30%|53.65 (-6.92)|65.14 (-2.70)|**66.72 (-1.76)**|
>
> We also added an anchor ablation.  The full NTS+STR model outperforms both the variant without anchor protection and NTS-only, which suggests that anchor protection contributes positively to final stability.
>
> |Setting|WAR|UAR|F1|
> |-|-|-|-|
> |NTS only|67.64|61.68|67.84|
> |NTS + STR w/o anchor protection |67.84|62.75|68.07|
> |NTS + STR (full)|**68.13**|**63.24**|**68.48**|

---

> > ### Author Rebuttal · Reviewer_UMAX · 2026-04-01
> >
> > Thank for your reply. I undertsand this is trianing-free approach but I really think world-mmodel is training-based, hence my criticism about this. I better understand your logic behind it, but still I do not agree with it. It is but a matter of opinion at this point, which is not relevant here.
> >
> > that said, thanks for the quantification of nts-str error propagation. I think this should be in the main paper (at least in summary and fully in appendix.)
> >
> > I update my score. I honestly think my original score was a bit unaligned to what I highlighted in the text. It should be weak reject first then I change it to weak accept due to the several strengths I outlined before, mainly the approach and hypothesis, now better accompagnied by ablation study on error propagation (which is still a flaw of the approach imo).
> >
> > Best

---

> > > ### Author Response · Authors · 2026-04-05
> > >
> > > Dear reviewer UMAX :
> > >
> > > We sincerely thank you for your positive assessment and for confirming that our rebuttal and additional experiments resolved your concerns. As suggested, we will faithfully incorporate the new results and clarifications into the final version of the manuscript. We appreciate your time and supportive guidance.

---

### Official Review · Reviewer_bikj · 2026-03-17

**Soundness:** 3
**Presentation:** 2
**Significance:** 4
**Originality:** 4
**Overall Recommendation:** 5
**Confidence:** 4

**Summary:**

The paper proposes a text-video emotion recognition using multimodal LLM. The key idea is to avoid fine-tuning the model. Instead, the approach increases the number of relevant tokens (STR) and reduces the influence of irrelevant tokens (NTS).

**Compliance With Llm Reviewing Policy:**

Affirmed.

**Final Justification:**

My score reflects my evaluation of this paper, which I find very interesting.

**Key Questions For Authors:**

How does this framework fit into the world-model formulation?

**Limitations:**

not discussed.

**Strengths And Weaknesses:**

Strenghs

The ideas in this paper are novel and interesting.

The paper provides an exhaustive evaluation, demonstrating the approach's strengths. It does include ablation analysis and parameter sensitivity.

Major limitations

I don’t see the world-based formulation in this paper. A world model is a system that represents entities (people, objects, concepts), captures relationships between them, and models how things change over time. I was expecting an internal model of how emotion is conveyed or perceived (e.g., appraisal theory). This is not the case. What is the world model that this paper is exploring? How does this framework fit into the world-model formulation?

The explanation of STR and NTS could be improved. It is hard to follow.

How many layers are “triggered”? “…during inference and admits the top-scoring tokens into K” & “Top-p emotion-relevant set construction.” How do you select how many tokens to affect? Does Figure 6 imply that you keep all the text tokens?

The analysis does not provide any statistical analysis to compare the results.

Minor limitations

“Rather than direct observations from multimoadl inputs.” Please correct the misspelling in the word multimodal.

Fix the grammar in this sentence: “Typical training-free methods such as text prompt
engineering to guide model behavior and improve outcomes without additional training, token merging to enhance processing efficiency by simplifying input data, and visual augmentation alongside multi-step inference to boost performance through visual enhancements and iterative processing.

This is about the sentence: “mix informative emotional evidence with modality-specific noise.” The word “noise” has a specific connotation in MER (e.g., occlusion, environmental noise, blurred image, etc.). In this context, does the word noise refer to information that is irrelevant for emotion recognition? If yes, you may want to clarify this.

Is the title in each subfigure in Figure 7 correct? The x-axis says visual keep ratio, while the title says “text keep radio =1.0” (radio?)

The results are more visually oriented than text-oriented. It would be nice to include some analysis of the text tokens (e.g., which text tokens are kept or removed).

---

> ### Author Rebuttal · Authors · 2026-03-31
>
> Dear Reviewer bikj:
>
> Thank you for your careful reading and constructive feedback. We address your concerns point by point below.
> # Q1: Meaning of the world-model-inspired view.
> We would like to clarify that our method is world-model-inspired, rather than a full world model. Here, this means a latent-state formulation of MER: emotion is treated as a latent state, and multimodal inputs as noisy observations. In the training-free setting, emotional cues are sparse, indirect, and buried in noisy context, making latent-state inference more suitable than direct feature matching. This view also goes beyond prompt engineering, heuristic token filtering, and direct feature aggregation by providing a principled way to distinguish evidence constraining the latent state from noise-dominated context. Under this view, tokens are observations with unequal contributions to latent-state inference, enabling systematic regulation of token admission and influence.
>
> As reflected in Eq. (1–8), our approach follows this formulation by modeling emotion as a latent state and refining token contributions as structured evidence for inference, consistent with latent-state world-model formulations [1–3]. In this world-model-inspired view, WETR instantiates a latent-state, inference-oriented view of MER and translates this principle into a practical inference-time token refinement mechanism for frozen MLLMs. This also explains the roles of the two modules: NTS determines which observations should enter inference, while STR regulates how strongly retained evidence contributes to latent-state estimation. We will revise the manuscript to clarify this scope and motivation.
>
> [1] Ha & Schmidhuber, World Models, NeurIPS 2018
>
> [2] Hafner et al., Dreamer, ICLR 2020
>
> [3] LeCun, A Path Towards Autonomous Machine Intelligence, 2022
> # Q2: Explanation of NTS and STR.
> We clarify that, under our world-model-inspired view, $z_e$ is not a physical-world entity, but the task-induced latent emotional state; video and text are noisy observations about that state; and tokens are the basic interface through which the model accesses such evidence. Accordingly, NTS acts as observation filtering, deciding which observations should enter state inference, while STR acts as evidence reweighting, deciding which retained cues should contribute more strongly to estimating the latent state. In this sense, NTS suppresses noise-dominated observations and keeps more informative evidence, while STR further strengthens complementary cues that provide stronger incremental constraints on the latent state. As a result, we believe that NTS/STR are not merely generic "selection+reweighting", but a simplified evidence-update mechanism organized around latent-state inference. We will make this correspondence much clearer in the revision.
> # Q3: Trigger layers, keep ratios, and text tokens.
> Thank you for pointing out the lack of implementation clarity. WETR is not applied to all layers. It is only activated at sparse trigger layers satisfying $l \bmod n_r=0$, while all other layers remain unchanged. For example, for InternVL3.5-8B (36 layers), we use $n_r=9$, so WETR is triggered at layers 9, 18, 27, and 36. We will make such backbone-specific trigger counts explicit in the revision.
>
> Regarding token selection, we do not keep all text tokens. We split text tokens into instruction tokens and text-content tokens. Only instruction tokens are always preserved for prompt stability and parsable outputs; text-content tokens are filtered by top-$p$ selection in the same way as visual tokens. Fig.3 already contains examples of dropped text tokens, and we will make this more explicit in the caption.
>
> For Fig.7, the reviewer is correct that the current presentation may be misleading. That figure shows the hyperparameter search process, not the final deployment setting. We first fixed the text keep ratio to 1.0 and searched the visual keep ratio, obtaining the best visual keep ratio of 0.03; we then fixed the visual keep ratio at 0.03 and searched the text keep ratio, obtaining the best text keep ratio of 0.2. Thus, the final setting is visual keep ratio=0.03 and text keep ratio=0.2, not keeping all text tokens.
> # Q4: Statistical comparison.
> Thank you for the suggestion. We have added statistical analysis. For MAFW, we report 5-fold mean±std. For CH-SIMS and MER23, we further perform paired significance tests based on test-set predictions.
>
> |Dataset|Method|WAR|UAR/Acc-3|F1|
> |-|-|-|-|-|
> |MAFW|Baseline (InternVL3.5-8B)|0.5508±0.0304|0.4421±0.0291|0.5419±0.0263|
> |MAFW|Ours|**0.5964±0.0298**|**0.4630±0.0276**|**0.5757±0.0225**|
>
> |Dataset|Test Type|WAR/Acc-2(p-value)|UAR/Acc-3(p-value)|F1(p-value)|
> |-|-|-|-|-|
> |CH-SIMS|Paired bootstrap|0.0132|0.0020|0.0108|
> |MER23|Paired bootstrap|< 0.001|0.0427|0.0007|
>
> # Q5: Text and figure issues.
> Thank you for these careful comments. We will correct the noted typos, grammar issues, figure labels, and ambiguous wording in the final version.

---

> > ### Author Rebuttal · Reviewer_bikj · 2026-04-03
> >
> > Thanks for the clarification. My score reflects my evaluation of this paper, which I find very interesting.

---

> > > ### Author Response · Authors · 2026-04-05
> > >
> > > Dear reviewer bikj :
> > >
> > > We sincerely thank you for your positive feedback and for confirming that your concerns are fully resolved. We appreciate your professional guidance throughout the process. We will ensure that all clarifications and additional analyses provided during the rebuttal are faithfully integrated into the final manuscript.

---

### Decision · Program_Chairs · 2026-04-30

**Decision:**

Accept (spotlight)

**Comment:**

This paper introduces WETR, a novel training-free method for Multimodal Emotion Recognition which refines token usage via noise suppression and state strengthening mechanisms. The reviewers consistently acknowledge the practical utility and strong empirical performance of the proposed method across multiple benchmarks. The primary point of contradiction was the concept of "world-model-inspired" framing. Reviewer EsuZ criticized this as lacking theoretical derivation, while the majority of reviewers accepted the author's clarification that this framing serves as a conceptual motivation based on latent-state inference rather than a strict implementation of a classical world model. The author's response during rebuttal stage addressed the concerns regarding prompt sensitivity, error propagation, and comparisons against other baseline methods. Given the solid technical contribution with consistent empirical gains, and the fact that the majority of reviewers vote for acceptance, I recommend acceptance.